# The Clinical Value of Pre-Diagnostic Thrombocytosis for the Detection of Lung Cancer in Primary Care

**DOI:** 10.3390/cancers16061154

**Published:** 2024-03-14

**Authors:** Melissa Barlow, Willie Hamilton, Sarah E. R. Bailey

**Affiliations:** Department of Health and Community Sciences, St. Lukes Campus, University of Exeter, Heavitree Road, Exeter EX1 2LU, UK; w.hamilton@exeter.ac.uk (W.H.); s.e.r.bailey@exeter.ac.uk (S.E.R.B.)

**Keywords:** platelet count, thrombocytosis, lung cancer, diagnosis, primary care

## Abstract

**Simple Summary:**

Thrombocytosis, an elevated platelet count, may suggest a patient has an undiagnosed lung cancer. This primary care study analysed data from English electronic medical records linked to the national cancer registry to understand how often thrombocytosis occurs before a lung cancer diagnosis, whether it is a feature of all the different subtypes of lung cancer, and its relation to the cancer’s stage at diagnosis. We found that the patients with lung cancer were nearly nine times more likely to have thrombocytosis before diagnosis compared to those without cancer. This association was strongest for squamous cell lung cancer. Thrombocytosis was linked to advanced stages of adenocarcinoma and squamous cell carcinoma, but it appeared equally in both the early and advanced stages of small-cell lung cancer.

**Abstract:**

Thrombocytosis is a risk marker for lung cancer in primary care. We investigated whether thrombocytosis presents pre-diagnostically for all the histological subtypes of lung cancer and its association with the stage at diagnosis. A matched cohort study used English electronic primary care data linked to the national cancer registry. Patients diagnosed with lung cancer aged ≥40 years with no prior history of malignancy were matched by age, sex, and general practice to five controls without lung cancer. Multivariable logistic regression models quantified the incidence of pre-diagnostic thrombocytosis and advanced-stage diagnoses, adjusting for COPD diagnosis, smoking status, and anti-platelet drug prescriptions. A total of 9504 cases were matched to 45,647 controls, consisting of 3260 (34%) adenocarcinomas (ADC), 2020 (21%) squamous cell carcinomas (SCC), 70 (<1%) large-cell carcinomas (LCC), and 1089 (12%) small-cell lung cancers (SCLC). The patients with lung cancer were 8.9 (95% CI 8.0–9.9) times more likely to exhibit pre-diagnostic thrombocytosis than the controls. The odds ratios were highest for the comparison between SCC and ADC (1.8, 95% CI 1.5–2.1). Thrombocytosis is associated with advanced-stage ADC and SCC but presented equally for early- and advanced-stage SCLC. Pre-diagnostic thrombocytosis may aid in the detection of all the histological subtypes in primary care.

## 1. Introduction

Lung cancer is the leading cause of cancer-related mortality, with 35,100 deaths annually in the UK alone [1]. Only 27% of patients with lung cancer are diagnosed at an early stage (stages I–II) [2]. The five-year survival rates range from 63% for patients diagnosed with stage I to only 4% for patients diagnosed at stage IV [2]. Due to the substantial influence of the stage at diagnosis on survival, the UK’s National Health Service has set a target to diagnose 75% of cancer patients at an early stage by 2028 [3].

The UK has recently announced the roll out of a national targeted lung cancer screening program in which patients who are aged from 55 to 74 years who have ever smoked will be invited for a lung health check [4]. Even with screening of high-risk groups, most lung cancers will continue to be diagnosed following symptomatic presentation to a primary care unit [5], and extensive research has focused on developing tools for facilitating earlier cancer detection in primary care. In recent years, thrombocytosis (a platelet count of ≥400 × 10^9^/L) has been recognised as a significant risk marker of lung cancer, which often precedes the onset of other symptoms and can be detected in primary care [6,7,8]. In recognition of its importance, the UK national guidance advises primary care clinicians to consider lung cancer in patients presenting with thrombocytosis [9].

There are multiple histological subtypes of lung cancer, and it is not known whether pre-diagnostic thrombocytosis is a generic feature of all lung cancers, or whether it is specific to certain subtypes. There are two main histological subtypes of lung cancer: non-small-cell lung cancer (NSCLC) and small-cell lung cancer (SCLC). NSCLC is further divided into adenocarcinoma (ADC), squamous cell carcinoma (SCC), large-cell carcinoma (LCC), and other rare subgroups [10]. SCLC is more aggressive than the NSCLC histologies [11], and only 5% of patients with SCLC are diagnosed at an early stage compared to approximately one-quarter of the patients with NSCLC [2,12]. Thus, using thrombocytosis as an entry route to earlier diagnosis may be particularly important in SCLC.

This study aimed to establish whether pre-diagnostic thrombocytosis was associated with all the histological subtypes of lung cancer, with secondary analysis evaluating its clinical value in the detection of early-stage lung cancer and whether this effect differed by histological subtype.

## 2. Methods

### 2.1. Data Sources and Patients

This was a matched cohort study using English electronic primary care data from the Clinical Practice Research Datalink (CPRD) GOLD [13] and Aurum [14], with a linkage to the National Cancer Registration and Analysis Service (NCRAS) [15]. Patients aged ≥40 years with no prior history of malignancy (excluding non-melanoma skin cancer) who were diagnosed with lung cancer between 1 January 2013 and 31 December 2016 were identified by the CPRD and each matched by age, sex, and general practice to five controls without lung cancer. The patients were classified into the following subtype groups dependent on their histological subtype diagnosis: ADC, SCC, LCC, or SCLC. Lung cancer cases (and their matched controls) with missing, ambiguous, or miscellaneous histology codes were not included in the subtype analyses.

### 2.2. Study Definitions

The lung cancer index date refers to the date of lung cancer diagnosis for each case; the same date was assigned to their matched controls. In accordance with the previous literature [6], thrombocytosis was defined as a platelet count of ≥400 × 10^9^/L blood, and pre-diagnostic refers to any time in the 365 days before their lung cancer index date. The patients without a platelet count record in the 365 days before their lung cancer index date were coded as not having pre-diagnostic thrombocytosis. Platelet counts of 0 and platelet counts ≥ 1000 × 10^9^/L were dropped from analysis. Smokers were defined as ever smokers if they had a smoking related medical code or a record of anti-smoking medication in the CPRD database before their lung cancer index date. Anti-platelet prescriptions included aspirin (75 mg only), clopidogrel, plavix, ticagrelor, brilinta, prasugrel, effient, dipyridamole, aggrenox, ticlopidine, ticlid, eptifibatide, or integrilin, and patients with a prescription record of one or more of these drugs in the 365 days before their lung cancer index date were categorised as being prescribed anti-platelet drugs. Chronic obstructive pulmonary disease (COPD) status was assigned if the patients had a record of COPD, emphysema, or chronic bronchitis in the CPRD before the date of their lung cancer diagnosis. The patients with lung cancer were classified as early stage if they were diagnosed at stages I-II or ‘limited stage’, and classified as advanced stage if they were diagnosed at stages III-IV or ‘extensive’ stage.

### 2.3. Sample Size Calculations

The incidence of thrombocytosis in the primary care population having a full blood count is 1.5–2.2% [16,17,18]. To have 90% power (alpha = 0.05) to detect a change in the prevalence of thrombocytosis from 1.5% to 5.0% for patients with lung cancer, a minimum of 289 patients for each lung cancer subtype was required. The histological subtypes of lung cancer are roughly 40% ADC, 30% SCC, 10% LCC, 3% other non-small-cell lung cancers, and 15% SCLC [19]. Therefore, to capture 289 patients with LCC (the rarest subtype), a minimum of 2890 total lung cancer cases were requested from the CPRD.

### 2.4. Statistical Analysis of Data

Stata’s ci prop command was used to calculate the binomial exact 95% confidence intervals (95% CIs) [20]. The proportion of lung cancer cases and their matched controls with pre-diagnostic thrombocytosis was reported with 95% confidence intervals. Conditional logistic regression, reflecting the matched design, estimated the odds ratios of pre-diagnostic thrombocytosis occurrence in lung cancer cases both unadjusted and adjusted for covariates. The covariates included smoking status (ever or never smoker), presence of an anti-platelet drug prescription (yes or no), and a diagnosis of COPD before their lung cancer diagnosis date (yes or no).

Multivariable logistic regression determined whether the histological subtype (categorised as ADC, SCC, LCC, or SCLC according to each patient’s histological subtype diagnosis) was an independent predictor of pre-diagnostic thrombocytosis status. If a subtype category had fewer than 289 cases (in accordance with sample size calculations), the patients diagnosed with this subtype were also excluded. The most prevalent subtype was used as the reference subtype. Covariates included sex (male or female), age group in 10-year bands (from 40 to 49, from 50 to 59, from 60 to 69, from 70 to 79, or ≥80 years), smoking status (ever or never smoker), the presence of an anti-platelet drug prescription (yes or no), a diagnosis of COPD before their lung cancer diagnosis date (yes or no), and stage of lung cancer at diagnosis (early or late).

A final multivariable logistic regression model evaluated whether the histological subtype and pre-diagnostic thrombocytosis status were independent predictors of early-stage diagnosis. This model also included an interaction term to assess whether the association of thrombocytosis and early-stage diagnosis differed by histological subtype. The additional covariates included sex (male or female), age group in 10-year bands (from 40 to 49, from 50 to 59, from 60 to 69, from 70 to 79, or ≥80 years), smoking status (ever- or never smoker), the presence of an anti-platelet drug prescription (yes or no), and a diagnosis of COPD before their lung cancer diagnosis date (yes or no).

For all regression analyses, backwards elimination and subsequent likelihood ratio testing determined which covariates were independent predictors of pre-diagnostic thrombocytosis occurrence or of advanced-stage disease using a *p*-value of ≤0.05 for retention. Finally, the marginal distributions of the models were used to obtain estimated incidences of pre-diagnostic thrombocytosis and advanced-stage diagnoses for lung patients adjusted for all the covariates. Sensitivity analysis were repeated using only the patients with a pre-diagnostic platelet count record.

All statistical analyses were performed on Stata v16.1 [21]. The results are reported in accordance with the Strengthening and Reporting of Observational Studies in Epidemiology (STROBE) statement [22] (Appendix A).

## 3. Results

### 3.1. Patient Characteristics

Data on 9821 patients with lung cancer were delivered from NCRAS. Of these, 180 were duplicates, and 137 had no CPRD data and were therefore excluded. After excluding 1989 controls due to missing patient identifiers or a case–control matching error in which they were included as both a case and a control, the remaining 9504 cases were matched to 45,647 controls (Figure 1).

The patient characteristics in the full dataset are outlined in Table 1. The median age of the patients was 72 years, and just over half were male (52.6%). The most common histological subtype was ADC, and one-third of the patients with lung cancer had ‘other’, missing, or ambiguous histology codes. Only 70 patients had an LCC histological diagnosis, so the patients with LCC were dropped from subtype analyses. Significantly more cases were classified as ever smokers, were diagnosed with COPD before their lung cancer index date, and had a platelet record in the year before their lung cancer index date. Almost all the patients had available data on the stage at diagnosis (95.2%), and of these, approximately one-quarter of the patients were diagnosed at an early stage.

The age and sex distribution of patients in the subset with a pre-diagnostic platelet count were similar, but had a slightly higher median age of 74 (IQR from 67 to 80) and consisted of slightly fewer males (51.2%).

### 3.2. Pre-Diagnostic Thrombocytosis

Table 2 shows the incidence of pre-diagnostic thrombocytosis for all the lung cancer cases and all the controls, with further stratification by histological subtype and sex. The incidence of pre-diagnostic thrombocytosis was much higher among the patients with lung cancer; this was 13.3% (95% CI 12.6 to 14.0%) compared to only 1.4% (95% CI 1.3 to 1.5%) of the matched controls. This effect was retained in the conditional logistic regression analyses when adjusting for smoking status, COPD diagnosis, and anti-platelet prescriptions. The adjusted odds ratio of pre-diagnostic thrombocytosis for the patients with lung cancer compared to that of the controls was 8.9 (95% CI 8.0 to 9.9, *p* < 0.001) (Appendix A).

The females had a higher proportion of pre-diagnostic thrombocytosis than the males, regardless of case–control status. Pre-diagnostic thrombocytosis was prevalent across all the subtypes of lung cancer; however, it was more prevalent among the patients with SCC than any other subtype.

The sensitivity analyses of all the patients with a platelet count yielded very similar results (Appendix A).

Table 3 outlines the adjusted odds ratios and the adjusted association of pre-diagnostic thrombocytosis from the multivariable logistic regression model. Histological subtype, sex, any previous COPD diagnosis, and age group data were all retained in the final multivariable logistic regression model as independent predictors of pre-diagnostic thrombocytosis. Smoking status and pre-diagnostic anti-platelet drug prescriptions were not significant predictors of thrombocytosis and were omitted from the final model. The adjusted odds ratios and incidence of pre-diagnostic thrombocytosis occurrence are outlined in Table 3. The patients with SCC were estimated to have had much higher rates of thrombocytosis than the ADC or SCLC patients, with no differences between ADC and SCLC. The females, younger patients, and patients with a previous diagnosis of COPD had a higher adjusted odds ratios and predicted incidences than their respective counterparts.

### 3.3. Stage at Diagnosis

A quarter (23.7%, 95% CI 22.9 to 26.7%) of the patients with lung cancer were diagnosed at an early stage. SCC was the subtype with the highest percentage diagnosed at an early stage (31.2%, 95% CI 29.1 to 33.3%), followed by ADC (26.6%, 95% CI 25.0 to 28.1%), with SCLC seldom diagnosed early (6.7%, 95% CI 5.2 to 8.3%). Of the patients with lung cancer with pre-diagnostic thrombocytosis, the proportion of patients diagnosed at an early stage was lower (14.7% (95% CI 12.8 to 16.8%)). This reduction was observed for the following patients with NSCLC histologies: 14.2% (95% CI 10.7 to 18.3%) of the patients with ADC with pre-diagnostic thrombocytosis and 20.1% (95% CI 16.6 to 25.5%) of the patients with SCC with pre-diagnostic thrombocytosis. However, for the patients with SCLC with pre-diagnostic thrombocytosis, there was no change in the proportion of patients who were diagnosed early, 7.1% (95% CI 3.3 to 13.0%).

These effects were largely unchanged in the presence of covariates (Table 4). Smoking status was the only covariate that was not an independent predictor of early-stage cancer. There was a significant interaction effect between pre-diagnostic thrombocytosis and the stage at diagnosis between SCLC and the reference subtype ADC (*p* < 0.05). This interaction was associated with the reduced incidence of early-stage ADC for the patients with pre-diagnostic thrombocytosis and an unchanged incidence of early-stage SCLC for the patients with pre-diagnostic thrombocytosis.

## 4. Discussion

This is the largest study exploring pre-diagnostic thrombocytosis occurrence in patients with lung cancer to date. Pre-diagnostic thrombocytosis was found to be a clinical feature of all the histological subtypes of lung cancer, with a higher prevalence among the patients with SCC, even after adjustment for the covariates. It is associated with advanced-stage NSCLC but presented equally among those with early- and late-stage SCLC.

These findings are supported by a previous systematic review of secondary care reports that found pre-treatment thrombocytosis to be a feature of all the histological subtypes of lung cancer [23]. However, the systematic review reported markedly higher proportions of pre-treatment thrombocytosis (27%). This difference probably represents the timing differences between the platelet counts of patients with lung cancer taken in the year before their diagnosis and in those later in their diagnostic/treatment pathways. Additionally, the studies included in this systematic review only included the patients with a platelet count record rather than assigning non-thrombocytosis status to the patients without a blood test, as in the present study [23]. Our sensitivity analysis including only the patients with a platelet count record in the year before their lung cancer index date found a higher rate of pre-diagnostic thrombocytosis: 18.3% (Appendix A). However, in a similarly designed study in primary care, the first to report thrombocytosis occurrence before a lung cancer diagnosis was made, just over a quarter (26%) of patients with lung cancer with an available platelet count measurement had pre-diagnostic thrombocytosis [18]. This may be partly due to the following subtype distribution: the study cohort favoured the SCC subtype, which we found to have more frequent thrombocytosis. Furthermore, with over a decade between the diagnosis dates of that study and of the patients in the present study, and continuing national efforts to diagnose cancers sooner, a higher proportion in the older study may have been diagnosed at an advanced stage. However, the stage at diagnosis was not reported, so this could not be confirmed [18].

A raised platelet count is recognised as a predictor of poorer survival among patients with lung cancer [24,25]. This may be related to the proportional rise in platelet count relative to the cancer’s growth. Future research should explore the specific timeframe during which a rise in platelet count precedes the diagnosis of lung cancer and whether assessing if a change in platelet count over time could offer a valuable tool in early lung cancer detection. A similar work evaluated this for bladder and renal cancer [26]. The observation that patients diagnosed with COPD in the present study are more often diagnosed at an earlier stage has also been reported recently in Canada and partly attributed to the higher utilisation of primary care [27]. Furthermore, uncovering the complex molecular mechanism responsible for elevating the platelet count in patients with lung cancer and establishing the cellular communication may provide a deeper insight into identifying novel diagnostic targets for earlier lung cancer detection.

While this study includes a representative cohort of the English lung cancer population, there are limitations that must be considered. Firstly, the reason for ordering the blood tests was not known. As more patients with lung cancer than the controls had a blood test, and the tested population in primary care have a higher risk of cancer [28], this may have influenced the results a little. However, it is unlikely that this affected our finding of higher prevalence across all the main histological subtypes, with additional prevalence in SCC, and the similar results reported by the sensitivity analysis using only the tested patients largely addresses this limitation. Furthermore, patients with infections or inflammation often exhibit elevated platelet counts [29], and patients with thrombocytosis may present to primary care generally more unwell than the patients with a normal platelet count, which may trigger GPs to consider further testing. A previous work compared cancer symptom presentation between a thrombocytosis cohort and a cohort of patients with a normal platelet count in primary care [6]. Overall, the difference in symptom presentation between the cohorts was minimal (<1%), other than for coughs, which were experienced by 3.9% of the thrombocytosis patients and 1.9% of the patients with a normal platelet count. Finally, while controlling for the most common co-morbidity in lung cancer (COPD), there may be other medical conditions that were not controlled for in the analyses that are more common in patients with lung cancer and may also cause an elevation in the platelet count [30].

## 5. Conclusions

In conclusion, pre-diagnostic thrombocytosis is a feature of all the main histological subtypes of lung cancer, and therefore may aid in the detection of all the histological subtypes in primary care. Our findings pertain to the patients in whom a platelet count is measured in primary care and in whom thrombocytosis has been identified. They do not refer to potential use as a screening instrument, which would require quite different research methodologies. In any case, it is unlikely thrombocytosis on its own could ever have sufficient discrimination to be useful in an asymptomatic screening population. Future research should determine the point in time in which the platelet count begins to rise in patients with lung cancer and assess whether a rise in platelet count could also be used as a predictor of undiagnosed lung cancer in primary care.

## Figures and Tables

**Figure 1 cancers-16-01154-f001:**
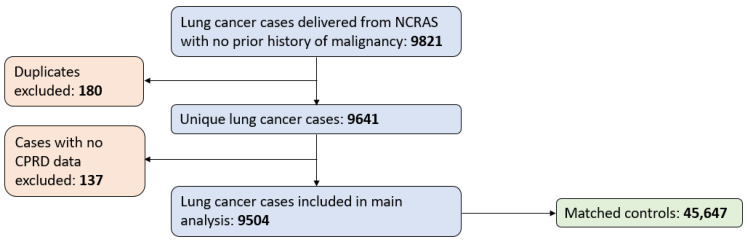
The patient selection process. There were 1740 controls received from the CPRD who were matched to themselves as a case, and 249 controls with a missing patient identifier. These were dropped before being matched to the lung cancer cases.

**Table 1 cancers-16-01154-t001:** Characteristics of study population.

		Lung Cancer Cases *n* = 9504			Matched Controls *n* = 45,647		
Median age (IQR)		72 (65 to 79)			72 (65 to 79)		
Male sex, %		52.6%			52.3%		
Histological subtype distribution, *n* (%)	ADC	3260 (34.3)					
	SCC	2020 (21.3)					
	LCC	70 (0.7)					
	SCLC	1098 (11.6)					
	Other or missing	3056 (32.2)					
		*n*	%	95% CI	*n*	%	95% CI
Diagnosed at an early stage		2149	23.7	22.9 to 24.7			
Ever smokers		9132	96.1	95.7 to 96.5	35,344	77.4	77.0 to 77.8
Anti-platelet prescriptions		3030	31.9	30.9 to 32.8	10,420	22.8	22.4 to 23.2
COPD diagnosis		2211	23.3	22.4 to 24.1	2041	4.5	4.3 to 4.7
Patients with a pre-diagnostic platelet record		6921	72.8	71.9 to 73.7	21,302	46.7	46.2 to 47.1

**Table 2 cancers-16-01154-t002:** Incidence of pre-diagnostic thrombocytosis in cases and controls categorised by sex and histological subtype. ADC, adenocarcinoma; SCC, squamous cell carcinoma; SCLC, small-cell lung cancer; CI, confidence interval.

	All Patients, % (95% CI)		Males, % (95% CI)		Females, % (95% CI)	
	Cases	Controls	Cases	Controls	Cases	Controls
All patients	13.3 (12.6 to 14.0)	1.4 (1.3 to 1.5)	11.1 (10.2 to 12.0)	0.8 (0.6 to 0.9)	15.8 (14.7 to 16.9)	2.0 (1.9 to 2.2)
ADC	10.7 (9.7 to 11.8)	1.3 (1.1 to 1.5)	8.6 (7.3 to 10.1)	0.7 (0.5 to 0.9)	12.3 (11.2 to 14.5)	1.9 (1.6 to 2.2)
SCC	16.9 (15.3 to 18.6)	1.2 (1.0 to 1.5)	15.1 (13.2 to 17.2)	0.8 (0.6 to 1.1)	19.9 (17.1 to 22.9)	1.9 (1.5 to 2.4)
SCLC	11.8 (10.0 to 13.9)	1.1 (0.8 to 1.5)	10.2 (7.8 to 13.1)	0.7 (0.4 to 1.1)	13.4 (10.7 to 16.6)	1.5 (1.1 to 2.1)

**Table 3 cancers-16-01154-t003:** Multivariable logistic regression for the predicted odds ratios and incidence of pre-diagnostic thrombocytosis occurrence for lung cancer cases only. Adjusted for the following significant predictors of pre-diagnostic thrombocytosis: subtype, sex, previous COPD diagnosis, and age group in 10-year bands. * baseline variable; OR, odds ratio; CI, confidence interval.

	Adj. OR	95% CI	*p*-Value	Adj. Incidence (%)	95% CI
Subtype					
ADC	1 *			10.8	9.7 to 11.8
SCC	1.7	1.5 to 2.0	<0.001	17.2	15.5 to 18.8
SCLC	1.1	0.9 to 1.3	0.6	11.4	9.5 to 13.2
Sex					
Male	1 *			11.0	10.0 to 12.0
Female	1.5	1.3 to 1.7	<0.001	15.1	13.8 to 16.4
Previous COPD diagnosis					
No	1 *			11.5	10.6 to 12.4
Yes	1.7	1.4 to 2.0	<0.001	17.9	15.8 to 19.9
Age group					
40 to 49	1 *			15.0	9.3 to 20.7
50 to 59	1.0	0.63 to 1.69	0.9	15.4	12.8 to 18.0
60 to 69	0.88	0.55 to 1.41	0.6	13.5	12.0 to 14.9
70 to 79	0.82	0.51 to 1.31	0.4	12.6	11.3 to 14.0
80+	0.63	0.38 to 1.03	0.06	10.0	8.2 to 11.9

**Table 4 cancers-16-01154-t004:** Multivariable logistic regression for the predicted odds ratios and incidence of an early-stage diagnosis for lung cancer cases only. Adjusted for thrombocytosis status, subtype, sex, age group, anti-platelet drug prescriptions (in the year before blood test), and previous COPD diagnosis. This model also included an interaction term for thrombocytosis status and subtype. * baseline variable; OR, odds ratio; CI confidence interval.

	Adj. OR	95% CI	*p*-Value	Adj. Incidence (%)	95% CI
Subtype					
ADC	1 *			26.4	24.9 to 28.0
SCC	1.3	1.1 to 1.4	0.001	31.6	29.5 to 33.7
SCLC	0.17	0.14 to 0.23	<0.001	6.5	5.1 to 8.0
Pre-diagnostic thrombocytosis status					
Normal	1 *			26.2	25.0 to 27.3
Thrombocytosis	0.40	0.29 to 0.55	<0.001	14.5	12.1 to 16.9
Interaction between thrombocytosis status and subtype					
Normal ADC				28.4	26.7 to 30.0
Thrombocytosis ADC				13.8	10.2 to 17.4
Normal SCC				33.4	31.1 to 35.7
Thrombocytosis SCC				20.0	15.9 to 24.3
Normal SCLC				6.6	5.0 to 8.1
Thrombocytosis SCLC				6.4	2.4 to 10.5
Sex					
Male	1 *			20.7	20.7 to 23.4
Female	1.4	1.2 to 1.6	<0.001	26.2	26.2 to 29.5
Age group					
40 to 49	1 *			18.9	12.7 to 25.0
50 to 59	1.3	0.82 to 2.0	0.26	22.9	19.9 to 25.9
60 to 69	1.5	0.95 to 2.2	0.09	24.9	23.1 to 26.7
70 to 79	1.5	0.98 to 2.3	0.07	25.4	23.7 to 27.2
80+	1.4	0.91 to 2.2	0.13	24.4	21.9 to 27.0
Previous COPD diagnosis					
No	1 *			23.2	22.1 to 24.4
Yes	1.4	1.2 to 1.6	<0.001	29.7	27.3 to 32.2
Anti-platelet drug prescription					
No	1 *			37.4	22.5 to 25.0
Yes	1.2	1.0 to 1.3	0.015	26.1	24.7 to 28.7

## Data Availability

Data were obtained from the CPRD with linkage to HES and NCRAS and were under license for this study. Therefore, restrictions may apply to the availability of these data.

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
