# Peer review of "The Clinical Value of Pre-Diagnostic Thrombocytosis for the Detection of Lung Cancer in Primary Care"

_cancers, 2024, doi:10.3390/cancers16061154_

Round 1

Reviewer 1 Report

Comments and Suggestions for Authors

Dear Editor and Authors,

It was my pleasure to review this manuscript titled “The clinical value of pre-diagnostic thrombocytosis for the detection of lung cancer in primary care” by Dr. Melissa Barlow and colleagues from the University of Exeter’s Department of Health and Community Sciences.

In this retrospective database analysis the authors attempt to establish a link between thrombocytosis and lung cancer so that it could be used as a screening tool in a primary care setting.

The hypothesis and basic premise of the study is somewhat over-reaching considering the fact that thrombocytosis could occur with a variety of conditions and therefore its specificity is variable/low.

Nevertheless, this is a well conducted study utilizing quite appropriate methodology and statistics. The authors have mined data from the Clinical Practice Research Datalink (CPRD) with linkage to the National Cancer Registration and Analysis Service (NCRAS) to conduct their analysis. They were thus able to gather data from over 55,000 patients total. Despite the sample been more than adequate they have performed a sample size calculation as is appropriate!! The authors have also utilized quite robust statistical analysis methods including propensity score matching.

They report that thrombocytosis incidence was higher in advanced stages  of adenocarcinoma and squamous cell carcinoma, but it was equal in both limited and extensive stages of small cell lung cancer.

As previously mentioned this is a well conducted study, it utilizes clear definitions, advanced statistical techniques and is well written and well presented in good and clear language.

My main real concern is with the premise of the study which suggests causality between lung cancer and thormocytosis! However, this link maybe associative and due to other factors and pathologies and not a direct result of the cancer. The authors need to put more effort in defending their premise by offering evidence from the literature regarding the existence of thrombocytosis with other forms of cancer and not focus specifically in lung cancer and primary care! Secondarily, their conclusion “In conclusion, pre-diagnostic thrombocytosis is a feature of all main histological subtypes of lung cancer and therefore may aid in the detection of all histological subtypes in primary care” is too bold a statement and too suggestive that thrombocytosis could be used as a screening tool!! I would therefore suggest some minor editing in the discussion including more data from the literature from other forms of cancer and also tone down their conclusions!! Apart from that, good job!

Comments on the Quality of English Language

Language is good as the authors are native speakers so only minor editing is required!

Author Response

Dear Editor,

Re: “The clinical value of pre-diagnostic thrombocytosis for the detection of lung cancer in primary care”

Thank you for sending the reviewers’ comments, which were largely positive. We have replied to the comments suggesting improvements or clarification but have removed responses to the (many) positive comments, as they did not require changes to the manuscript. Our response to reviewer 1 is attached.

We also attach a revised manuscript in which changes have been highlighted yellow. In addition to the amendments we have made based on the reviewers comments, we have also provided additional citations to meet Cancer’s minimum number of references. The new citations have also been highlighted in the revised manuscript.

Kind regards,

Melissa Barlow

Reviewer 2 Report

Comments and Suggestions for Authors

The study is well done. However, there are two major concerns:

• it is essentially repetitive research by the same author and other authors. Without adding new important information.

Bailey SE, Ukoumunne OC, Shephard EA, Hamilton W. Clinical relevance of thrombocytosis in primary care: a prospective cohort study of cancer incidence using English electronic medical records and cancer registry data. Br J Gen Pract. 2017 Jun;67(659):e405-e413. doi: 10.3399/bjgp17X691109. Erratum in: Br J Gen Pract. 2021 Sep 30;71(711):445. PMID: 28533199; PMCID: PMC5442956.

Giannakeas V, Narod SA. Incidence of Cancer Among Adults With Thrombocytosis in Ontario, Canada. JAMA Netw Open. 2021 Aug 2;4(8):e2120633. doi: 10.1001/jamanetworkopen.2021.20633. PMID: 34383058.

Giannakeas V, Kotsopoulos J, Cheung MC, Rosella L, Brooks JD, Lipscombe L, Akbari MR, Austin PC, Narod SA. Analysis of Platelet Count and New Cancer Diagnosis Over a 10-Year Period. JAMA Netw Open. 2022 Jan 4;5(1):e2141633. doi: 10.1001/jamanetworkopen.2021.41633. PMID: 35015064; PMCID: PMC8753503.

• The main aim of the study („This study aimed to establish whether pre-diagnostic thrombocytosis was associated with all histological subtypes of lung cancer, with a secondary analysis evaluating its clinical value in the detection of early-stage lung cancer, and whether this effect differed by histological subtype“) has no practical meaning or importance.

If there is any suspicion of cancer, such a person must be examined for cancer. Histological verification is necessary to confirm cancer. During histological verification, the cancer subtype is also known. In any case, there is no room for speculation or assumptions in this situation by thrombocytosis.

Other concerns:

• The statement in lines 36-37: "Only 27% of lung cancer patients are diagnosed at an early stage (stage III-IV)" is incorrect.

• The statement in lines 41-43: "Even with screening of high-risk groups, most lung cancers are diagnosed following symptomatic presentation to primary care, and extensive research has focused on developing tools for early detection before the onset of symptoms" is unclear, partly incorrect. Screening programs for lung cancer (as with other cancers) are performed on asymptomatic individuals. Patients with clinical symptoms are subject to a targeted examination. In both cases, the methods are different.

Author Response

Dear Editor,

Re: “The clinical value of pre-diagnostic thrombocytosis for the detection of lung cancer in primary care”

Thank you for sending the reviewers’ comments, which were largely positive. We have replied to the comments suggesting improvements or clarification but have removed responses to the (many) positive comments, as they did not require changes to the manuscript. Our response to reviewer 2 is attached.

We also attach a revised manuscript in which changes have been highlighted yellow. In addition to the amendments we have made based on the reviewers comments, we have also provided additional citations to meet Cancer’s minimum number of references. The new citations have also been highlighted in the revised manuscript.

Kind regards,

Melissa Barlow

Round 2

Reviewer 2 Report

Comments and Suggestions for Authors

None